# Simulating bout-and-pause patterns with reinforcement learning

**Kota Yamada**[1,2]*, **Atsunori Kanemura**[2]

**1** Keio University, Tokyo, Japan, **2** LeapMind Inc. Tokyo, Japan

* haroldthebarrel.yk@gmail.com

## Abstract

Animal responses occur according to a specific temporal structure composed of two states, where a bout is followed by a long pause until the next bout. Such a bout-and-pause pattern has three components: the bout length, the within-bout response rate, and the bout initiation rate. Previous studies have investigated how these three components are affected by experimental manipulations. However, it remains unknown what underlying mechanisms cause bout-and-pause patterns. In this article, we propose two mechanisms and examine computational models developed based on reinforcement learning. The model is characterized by two mechanisms. The first mechanism is choice—an agent makes a choice between operant and other behaviors. The second mechanism is cost—a cost is associated with the changeover of behaviors. These two mechanisms are extracted from past experimental findings. Simulation results suggested that both the choice and cost mechanisms are required to generate bout-and-pause patterns and if either of them is knocked out, the model does not generate bout-and-pause patterns. We further analyzed the proposed model and found that it reproduced the relationships between experimental manipulations and the three components that have been reported by previous studies. In addition, we showed alternative models can generate bout-and-pause patterns as long as they implement the two mechanisms.

## Introduction

Animals engage in various activities in their daily lives. For humans, they may be working, studying, practicing sports, or playing video games. For rats, they may be grooming, foraging, or escaping from a predator. Although specific activities are different between different species, common behavioral features are often observed.

Bout-and-pause patterns are one of the behavioral features commonly observed in many species. Activities engaged by an animal do not occur uniformly through time but often have short periods in which a burst of engaged responses is observed. For example, in an operant conditioning experiment, a rat presses a lever repeatedly in a short period and then it stops lever pressing. After a moment, the rat starts lever pressing again. The rat switches between the lever pressing behavior and the no lever pressing behavior again and again throughout the experiment. Such a temporal structure comprising of short-period response bursts and long

**Data Availability Statement:** Simulation programs are available from GitHub: https://github.com/echo0yasum1/simulating_bout_and_pause_pattern.

**Funding:** This study was supported in part by Grant-in-Aid for JSPS Fellows (20J21568) to KY from the Japan Society for the Promotion of Science (http://www.jsps.go.jp/english/e-grants). The funder had no role in study design, data collection, data analysis, and preparation of the manuscript. KY and AK are employed by and receive salaries from LeapMind Inc. (https://leapmind.io/en/), and the both authors played roles in the study design, data collection and analysis, decision to publish, or preparation of the manuscript. The specific roles of these authors are articulated in the 'author contributions' section. There was no additional external funding received for this study.

**Competing interests:** KY and AK are employed by and receive salaries from LeapMind Inc. This does not alter our adherence to PLOS ONE policies on sharing data and materials. There are no patents, products in development, or marketed products to declare.

pauses is observed in various species and activities; for example, email and letter communication by humans [1], foraging by cows [2], and walking by Drosophila [3].

Shull et al. [4] showed that bout-and-pause patterns, observed under an environment where rewards are available probabilistically at a constant rate (variable interval (VI) schedule), can be described with a broken-stick shape in the log-survivor plot of interresponse times (IRTs), which are characterized by a bi-exponential probability model. If IRTs follow a single exponential distribution, then the log-survivor plot shows a straight line. If IRTs follow a mixture exponential distribution called a bi-exponential model, the log-survior plot shows a broken-stick shape composed of two straight lines that have different slopes. Killeen et al. [5] found that lever pressing by rats is well described with a bi-exponential model, suggesting that this behavior has a bout-and-pause pattern. If IRTs follow a bi-exponential distribution, there are two different types of responses; within-bout responses, which have short IRTs, and between-bout responses, which have long IRTs. Each response type has its own exponential distribution in a bi-exponential model. Killeen et al. [5] formulated the bi-exponential model as follows:

$$p(IRT = \tau) = (1 - q)\omega e^{-\omega\tau} + qbe^{-b\tau}, \tag{1}$$

where the first term describe IRTs of within-bout responses and the second term describes IRTs of between-bout responses. This model has three free parameters: $q$, $\omega$, and $b$, each of which corresponds to a different component in bout-and-pause patterns. First, $q$ denotes the mixture ratio of the two exponential distributions in the model and it corresponds to the mean length of a bout. The bout length is the number of responses contained in one bout. Second, $\omega$ denotes the rate parameter for the exponential distribution of within-bout IRTs and it corresponds to the within-bout response rate. Finally, $b$ denotes the rate parameter for the exponential distribution of between-bout IRTs and it corresponds to the bout initiation rate. These three model parameters define the overall response rate. They are also called bout components.

The bout length, the within-bout response rate, and the bout initiation rate are affected by motivational and schedule-type manipulations [4, 6–11]. Motivational manipulations include the reinforcement rate, the response-reinforcement contingency, and the deprivation level. An example of schedule-type manipulations is adding a small variable ratio (VR) schedule in tandem to a variable interval (VI) schedule.

Table 1 summarizes existing findings on the relationships between experimental manipulations and the two bout components. The bout length was reported to be affected by manipulations as follows:

- It increases or stays the same as the reinforcement rate increases [4, 6].

- It increases or stays the same as the deprivation level increases [4, 7, 8].

- It decreases or stays the same by extinction [10, 12].

- It increases by tandem VR [4, 6, 13]. When a VI schedule is followed by a small VR (tandem VI VR), an animal stays in a bout longer and emits more responses in each bout.

The bout initiation rate was reported to be:

- It increases as the reinforcement rate increases [4, 6, 14, 15].

- It increases as the deprivation level increases [4, 7, 8].

- It decreases by extinction [10, 12, 16].

**Table 1. Previous findings from animal experiments on the relationships between manipulations and bout components.**

| | Motivational | | | Schedule type |
|---|---|---|---|---|
| | **Reinforcement rate** | **Deprivation level** | **Extinction** | **Tandem VR** |
| Bout length | ↗ or →? | ↗ or →? | ↘ or →? | ↗ |
| Bout initiation rate | ↗ | ↗ | ↘ | ↘ or →? |

The three cells marked with "?" do not have agreement within the previous reports.

- It decreases or stays the same by tandem VR [10]. Brackney et al. [10] showed that if we add a small VR schedule in tandem to a VI schedule, the bout initiation rate decreased slightly.

Although the previous studies have investigated the relationships between some experimental manipulations and the bout components, we still do not know how to construct a model that generate bout-and-pause patterns based on the experimental findings. Smith et al. [17] showed experimentally that choice and cost play important roles in organizing responses into bout-and-pause patterns. When pigeons were trained under a single schedule, the log-survivor plot did not show a broken-stick shape [18, 19]. Smith et al. [17] trained pigeons under a concurrent VI VI schedule with and without a changeover delay (COD). When pigeons were trained under the concurrent VI VI schedule without a COD, the log-survivor plot still did not show a broken stick, resulting in the a straight line. However, under the concurrent VI VI schedule with a COD, the log-survivor plot showed a broken stick, indicating that bout-and-pause patterns were clearly observed. Similar observations have been made for rats, assuming that they engage in alternative behaviors during conditioning [20]. From these experimental observations, we extracted the following three facts. 1) When animals engage only in one response in a given situation, bout-and-pause patterns are not observed. 2) If animals can choose responses from two alternatives without a COD, bout-and-pause patterns are still not observed. 3) Considering 1) and 2), we conclude that bout-and-pause patterns are organized only when animals have two (or more) possible alternatives under a given situation (i.e., choice is available) and there is a COD between the start of engagement and a reinforcement (i.e., cost is associated with a changeover). These facts are interesting but they remain to be inductive and we still do not have constructive explanation that generate bout-and-pause patterns. Existing studies on bout-and-pause patterns have investigated to describe the phenomena rather than to provide constructive models. Although many models have been proposed [5, 9, 21]), they are descriptive and did not answer the question of "what mechanisms shape responses into bout-and-pause patterns?"

Kulubekova and McDowell [22] examined a computational model aimed to reproduce bout-and-pause patterns based on the principle of selection by consequences developed by McDowell [23] but they did not test which mechanisms are behind bout-and-pause patterns. In other words, they showed that a computational model of selection by consequence could reproduce bout-and-pause patterns but did not show minimal requirements to reproduce them.

In this article, we propose a computational model based on reinforcement learning that accounts for the constructive mechanism of bout-and-pause patterns. We assume that bout-and-pause patterns are generated by two mechanisms: a choice between operant and other behaviors and a cost that is required to a transition from one behavior to another. We suppose that motivational manipulations affect only the choice mechanism and schedule-type manipulations affect the cost mechanism. To incorporate those two mechanisms, we design a three-state Markov transition model, which has an extra state in addition to the bout and pause

states. We perform three simulation studies to analyze the proposed model. In Simulation 1, we introduce our model on the basis of the two different mechanisms, choice and cost. We show that the proposed model can reproduce bout-and-pause patterns by finding that the log-survivor plot shows a broken-stick shape. We compare three models: a dual model, a no cost model, and a no choice model. The dual model is composed of both the choice and cost mechanisms. The no cost model has only the choice mechanism and the no choice model has only the cost mechanism. Simulation results demonstrate that the dual model can reproduce bout-and-pause patterns but the other two models failed to reproduce them. It implies that both choice and cost are required for animal responses to be organized into bout-and-pause patterns. In Simulation 2, we analyze the dual model in depth and report its behavior under various experimental settings to test if the dual model can reproduce the relationships between the experimental manipulations and the bout components discovered so far. Simulation results suggest that the dual model can reproduce them not only qualitatively but also quantitatively. In Simulation 3, we show that a two-state model can also reproduce bout-and-pause patterns even without the third state because it incorporates the two mechanisms. However, having the third state is useful for separating the effects of the choice and cost mechanisms. We speculate that real animals might have similar mechanisms that generate bout-and-pause patterns as the dual model, which can be a useful computational tool for studying animal behavior.

# 1 Simulation 1

## Material and method

**Model.**   Our model is based on reinforcement learning [24]. We designed a three-state Markov process for modeling bout-and-pause patterns (Fig 1(a)). Two of the three states are "Operant" and "Others," in which the agent engages in the operant behavior or performs other behaviors, respectively. We call them Operant and Others instead of engagement or visit and disengagement or pause, thereby we emphasize that bout-and-pause patterns are results of a choice between the operant and other behaviors. In the third "Choice" state, the agent makes a decision between the operant and other behaviors. By having the Choice state in our model, we incorporate the knowledge that animals can choose their behavior from available options (e.g. grooming, exploration, and excretion) when they move freely during an experiment. The second knowledge is a cost required to make a transition from one behavior to another. Animals must decide whether to keep doing the same behavior or to make a transition, because a fast switching is not optimal if a transition incurs a cost. Fig 1(b) and 1(c) shows two knockout models, the no choice model and the no cost model, respectively. In each model, one of the two mechanisms from the dual model is removed. In the no choice model, an agent can choose only the operant behavior in a given situation. In the no cost model, no cost is required when a transition is made.

Here is how the agent travels in the proposed model. In the Choice state, the agent chooses either the operant or other behaviors. As a result of the choice, it moves from the Choice state to one of the Operant or Others states. It makes the choice based on the preference for each behavior, which is denoted by $Q_{\mathrm{pref}}$. We will explain how to calculate $Q_{\mathrm{pref}}$ in the next paragraph. In the Operant state, the agent engages in the operant behavior, and, after every response, it decides whether to stay in the Operant state or to move back to the Choice state. It decides to stay or move based on $Q_{\mathrm{cost}}$, which represents a transition cost to the Choice state, whose mathematical definition will be given later in this Model section. The Others state is the same as the Operant state except for that the agent performs other behaviors.

The preference $Q_{\mathrm{pref}}$ is a function that compares the operant and other behaviors when the agent makes a choice between them. The $Q_{\mathrm{pref}}$ function changes over time since it is updated

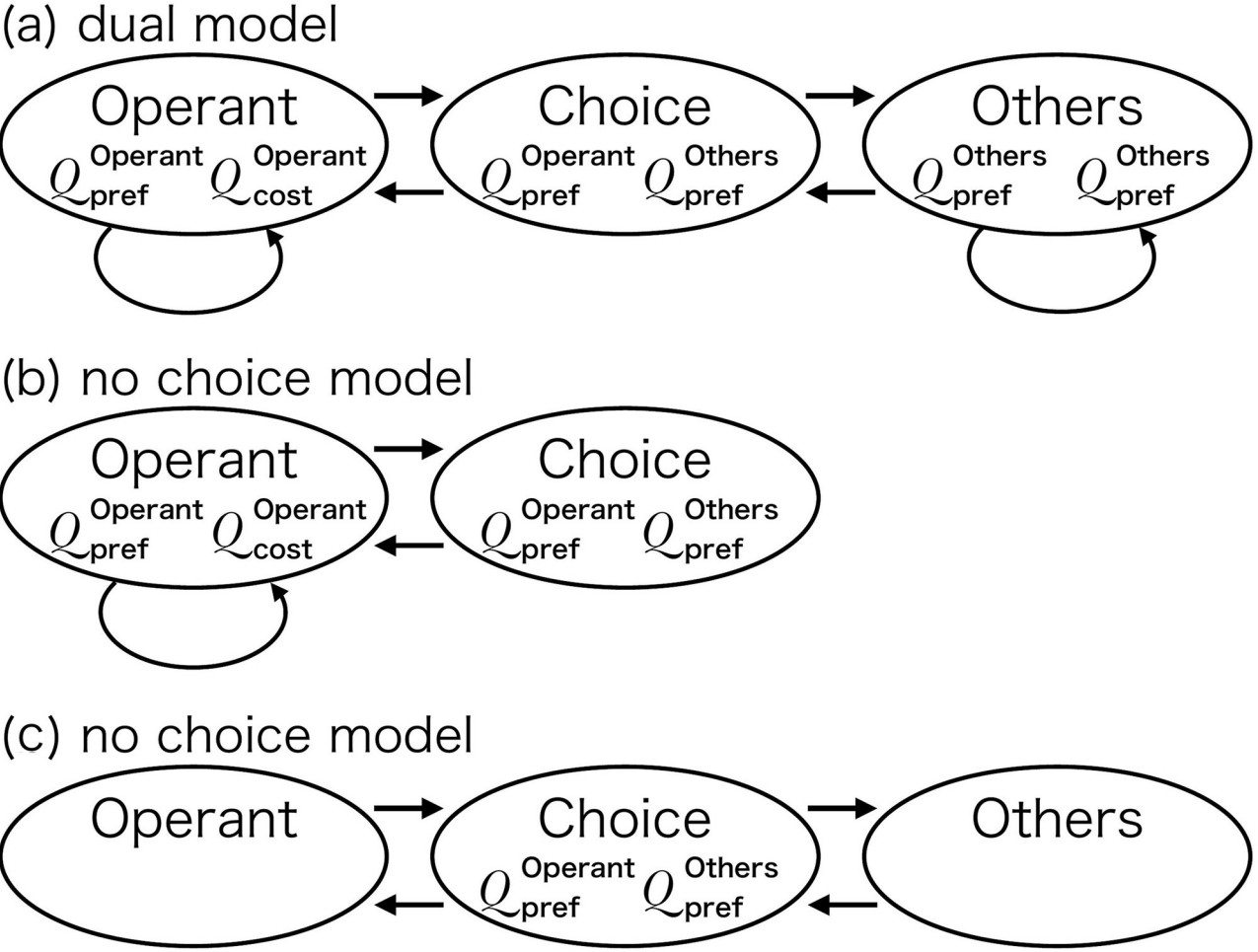

**Fig 1. Model schemes of the dual model, the no choice model and the no cost model.** (a) The model scheme of the dual model. The upper node, the bottom left node, and the bottom right node correspond to the Choice state, the Operant state, and the Others state respectively. Each arrow denotes the transition from one state to others. (b) The model scheme of the no choice model. In this model, the Others state is omitted. (c) The model scheme of the no cost model. In this model, self-transitions in the Operant and Others state is omitted.

based on the presence (or absence) of a reinforcer per bout. The following equation describes the updating rule for $Q_{pref}$:

$$Q_{pref}^{(i)}(t+1) = \begin{cases} Q_{pref}^{(i)}(t) + \alpha_{rft}(r^{(i)}(t) - Q_{pref}^{(i)}(t)), & \text{if a reinforcer is presented,} \quad (2a) \\ Q_{pref}^{(i)}(t) + \alpha_{ext}(0 - Q_{pref}^{(i)}(t)), & \text{otherwise,} \quad (2b) \end{cases}$$

where $t$ denotes time in session; $\alpha_{rft}$ and $\alpha_{ext}$ denotes the learning rates of reinforcement and extinction, respectively; $r$ denotes the reinforcer value and we assume $r > 0$ when a reinforcer is present and $r = 0$ when a reinforcer is absent; and $i \in \{\text{Operant, Others}\}$ denotes each option, that is, $i = \text{Operant}$ if the operant behavior is chosen and if $i = \text{Others}$ if other behaviors are chosen. We omit superscript $(i)$ and denote $Q_{pref}$ when it can be any of $i = \text{Operant}$ or Others.

In the Choice state, the agent chooses either of the Operant or Others states according to the probability distribution calculated from the preferences for the two behaviors. The

probability of transition to option $i \in \{$Operant, Others$\}$ is defined as follows:

$$p_i = \frac{\exp\{\beta Q_{\text{pref}}^{(i)}(t)\}}{\sum_{i \in \{\text{Operant,Others}\}} \exp\{\beta Q_{\text{pref}}^{(i)}(t)\}}, \tag{3}$$

where the softmax inverse temperature parameter $\beta$ represents the degree to which a choice is focused on the highest-value option.

The cost $Q_{\text{cost}}$ is a function that defines a barrier in making a transition from the performed behavior to the Choice state. We assumed that the cost is independent from the preference and depends only on the number of responses that are emitted to obtain a reinforcer from a bout initiation. When a reinforcer is presented, the cost function $Q_{\text{cost}}$ is updated according to

$$Q_{\text{cost}}^{(i)}(t+1) = Q_{\text{cost}}^{(i)}(t) + \alpha_{\text{rft}}(\log x^{(i)}(t) - Q_{\text{cost}}^{(i)}(t)), \tag{4}$$

where $x$ denotes the number of responses that are emitted to obtain a reinforcer in a bout. Then, $x$ is initialized to 1 when the agent receives a reinforcer or comes back to the Choice state without a reinforcer. The other parameters are the same as Eqs (2a) and (2b). The same ($i$)-omitting rule applies also to $Q_{\text{cost}}$. In Eq (4), $x$ is attenuated by taking its logarithm. This is because, if we do not attenuate $x$, the barrier defined by $Q_{\text{cost}}$ becomes too high and the agent keeps staying at the performed state. To avoid it, we employed Fechner's law [25] to make the performed state less attractive.

If the agent is in either of the Operant or Others states, it makes a decision whether to stay in the same state or to go back to the Choice state. A decision is made according to the probability of staying in the same state calculated from the cost and the preference for the state, which is defined as follows:

$$p_{\text{stay}}^{(i)} = \exp\left\{\frac{-1}{w_{\text{pref}}Q_{\text{pref}}^{(i)}(t) + w_{\text{cost}}Q_{\text{cost}}^{(i)}(t)}\right\}, \tag{5}$$

where $w_{\text{pref}}$ and $w_{\text{cost}}$ are positive weighting parameters for $Q_{\text{pref}}$ and $Q_{\text{cost}}$, respectively. We assumed $w_{\text{cost}} > w_{\text{pref}}$ because schedule-type operations have stronger effects on the bout length than motivational manipulations. When $Q_{\text{pref}}$ or $Q_{\text{cost}}$ increase, $p_{\text{stay}}$ increases too.

**Simulation.** In Simulation 1, we compared the three possible models; the dual model, the no choice model, and the no cost model. The dual model (Fig 1(a)) includes both the choice and cost mechanisms as we described in the Model section. The second model was the no choice model (Fig 1(b)), which has only the cost mechanism and it can be thought of as a model made by removing the choice mechanism from the dual model. In the no choice model, the agent only engages in the operant behavior. In other words, this model chooses only the operant behavior in the Choice state. The third model was the no cost model, which has only the choice mechanism without the cost mechanism. The no cost model chooses either operant or other behavior independent of the previous behavior; that is, according to this model, the agent does not continue to be in the same state and comes back to the Choice state after each response. In the no cost model, the self transition paths were removed because $p_{\text{stay}}$ is very low without having $Q_{\text{cost}}$ in Eq (5).

Simulation conditions were as follows. The schedule for the operant behavior was VI 120 s (0.5 reinforcer per min) without an inter-trial interval, and the schedule for the other behavior was FR 1. The maximum number of reinforcers in the Operant state was 1,000; that is, if the number of reinforcers reached 1,000, the simulation was terminated. The value of a reinforcer given by taking the operant behavior was $r^{(\text{Operant})} = 1.0$ and that by taking other behaviors was $r^{(\text{Others})} = 0.5$. The model parameters were $\alpha_{\text{rft}}$, $\alpha_{\text{ext}}$, $\beta$, $w_{\text{pref}}$, and $w_{\text{cost}}$. We set $\alpha_{\text{rft}} = 0.05$,

$\alpha_{\text{ext}} = 0.01$, $\beta = 12.5$, $w_{\text{pref}} = 1.0$ and $w_{\text{cost}} = 3.5$. The response probabilities in the Operant and the Others states were fixed at 1/3 in each time step. These parameters were designed based on the knowledge on experimental conditions, e.g., the reinforcer for the operant behavior should be higher than that for other behaviors, implying $r^{(\text{Operant})} > r^{(\text{Others})}$. Before the start of the simulation, we initialized the agent and the experimental environment. The initial values of $Q_{\text{pref}}^{(i)}$ and $Q_{\text{cost}}^{(i)}$ were both 0 and we created a VI table according to Flesher and Hoffman [26]. We set the time step in the simulation to be 0.1 s.

We show pseudocode of the model and simulation in Algorithm 1, where *NumResponses* means *x* in Eq 4 and the three *Behavior()* functions are defined in Algorithms 2, 3, and 4. We implemented the algorithm in Julia 1.0 and ran simulations on a computer with a 1.80 GHz Intel i7-8565 processor, 16 GB of RAM, and 1 TB of SSD, operating with Ubuntu 18.04 LTS. The same configuration was used also for Simulations 2 and 3. The Julia code is available at: https://github.com/echo0yasum1/simulating_bout_and_pause_pattern.

**Algorithm 1** Pseudocode of simulation

```
t ← 0, NumRewards ← 0, ResponseTimes ← {}, i ← Choice
while NumRewards < 1000 do
  t ← t + 0.1
  if i = Choice then
    ChoiceBehavior()
  end if
  if i = Operant and uniform(0, 1)≤1/3 then
    OperantBehavior()
  end if
  if i = Others and uniform(0, 1)≤1/3 then
    OthersBehavior()
  end if
end while
```

**Algorithm 2** Definition of *ChoiceBehavior()*

```
Select a state i ∈ {Operant, Others} with probability defined by Eq (3)
NumResponses ← 1
```

**Algorithm 3** Definition of *OperantBehavior()*

```
Append t to ResponseTimes
NumResponses ← NumResponses + 1
Select a state i ∈ {Operant, Choice} with probability defined by Eq (5)
if reward is presented then
  Update Q_pref^(Operant)(t) according to Eq (2a)
  Update Q_cost^(Operant)(t) according to Eq (4)
  NumRewards ← NumRewards + 1
  NumResponses ← 1
end if
if reward is absent then
  if i = Choice then
    Update Q_pref^(Operant)(t) according to Eq 2b
  end if
end if
```

**Algorithm 4** Definition of *OthersBehavior()*

```
NumResponses ← NumResponses + 1
Update Q_pref^(Others)(t) according to Eq (2a)
Update Q_cost^(Others)(t) according to Eq (4)
reward is presented according to FR 1
NumResponses ← 1
Select a state i ∈ {Others, Choice} with probability defined by Eq (5)
```

## Results: Simulation 1

Fig 2(a) shows event records of IRTs generated by each model and Fig 3 shows the model schemes with transition probabilities. The top panel of Fig 2(a) shows that the no choice model generated a dense repetition of only the operant behavior at a high rate without long pauses. From Fig 3, the probability the agent stayed in the Operant state was empirically 0.95. In the middle panel of Fig 2(a), the response rate under the no cost model was low and each response was separated by long pauses. From Fig 3, the probability of the agent choosing to transit to the Operant state was empirically 0.06 and the agent returned to the Choice state immediately after it responded. In the bottom panel of Fig 2(a), the agent with the dual model generated a repetitive pattern of responses with a high rate in a short period followed by a long pause. From Fig 3, the agent in the Choice state made a transition to the Operant state with a 0.12 probability and it stayed in the Operant state with a 0.71 probability.

Fig 2(b) show log-survivor plots to see whether they show a straight line or a broken stick. We used the IRTs from after the agent obtained 500 reinforcers to the end of the simulation. The log-survivor plots of the no choice model and the no cost model were described by one straight line whereas that of the dual model was described with a broken-stick shape. The no choice model has a steeper slope than the no cost model and is tangential to the curve of the dual model at the leftmost position. The no cost model slope was slightly steeper than that of the dual model at the right side.

### 1.1 Discussion: Simulation 1

Both the event records and log-survivor plots in Fig 2 imply that only the dual model generated bout-and-pause patterns and the other two models failed to reproduce bout-and-pause patterns. The event records in Fig 2(a) suggests that only the dual model exhibit bout-and-pause patterns. The log-survivor plot of only the dual model in Fig 2(b) showed not a straight but a

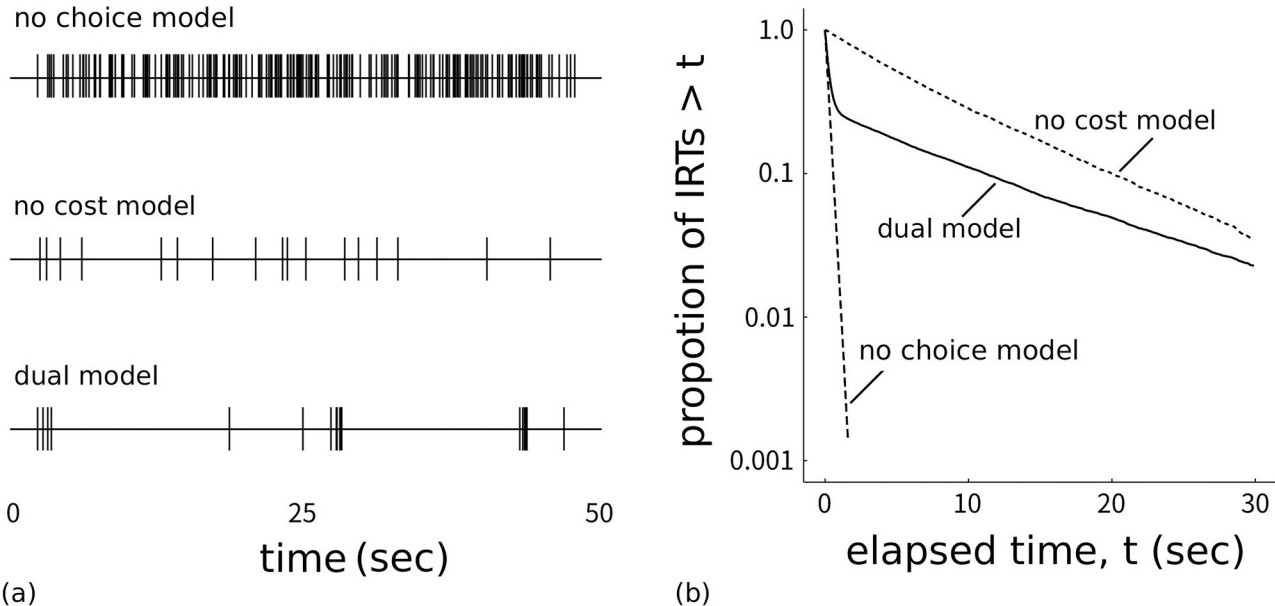

**Fig 2.** (a) Response event records in the (top) no choice, (middle) no cost, and (bottom) dual models in the 50 s period just after 500 reinforcers were presented (event records were stable after 500 reinforcer presentations). Each vertical line denotes one response. (b) Log-survivor plots of the three models drawn by using all the IRTs after 500 reinforcers.

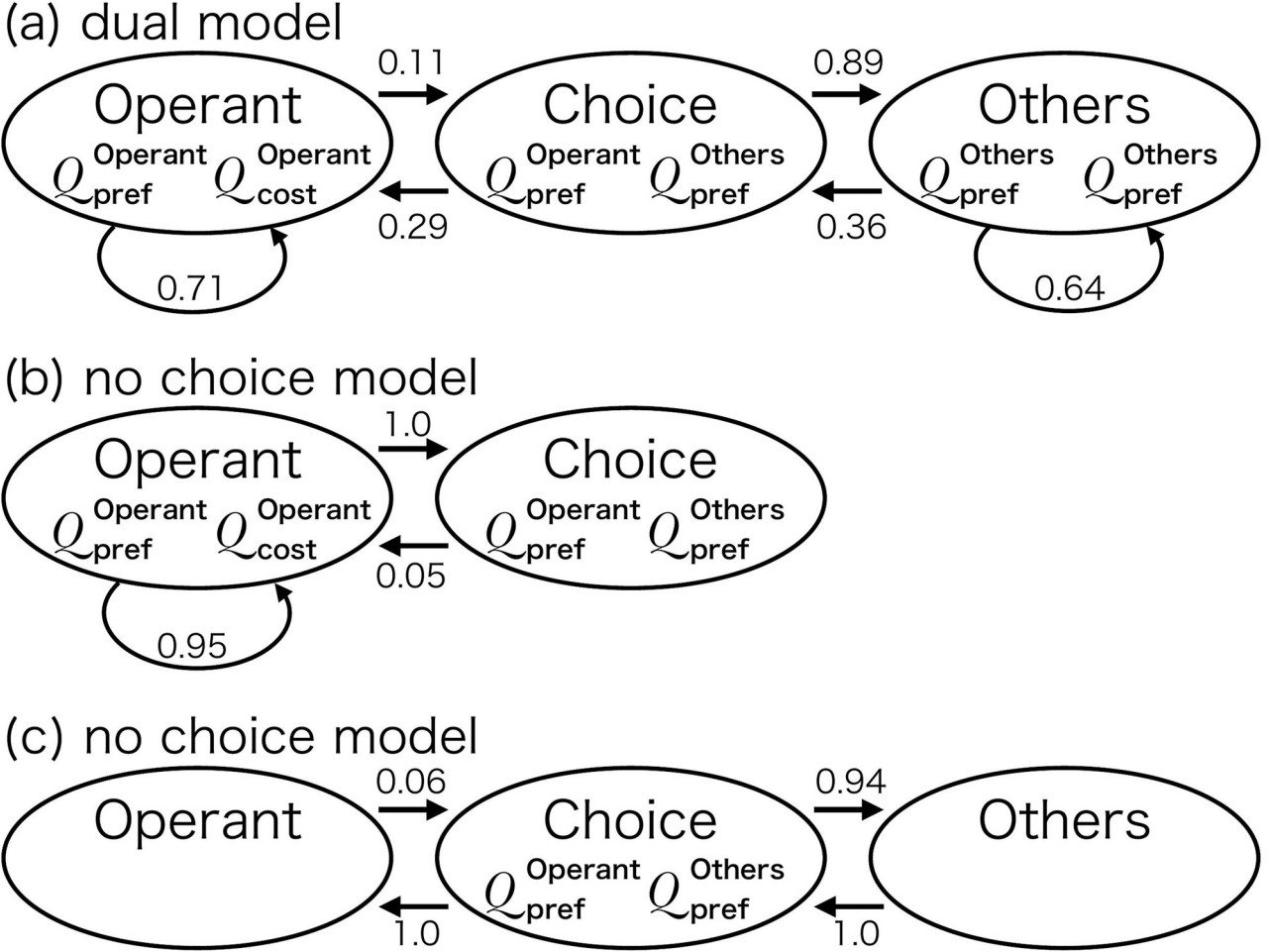

**Fig 3. The transition probabilities between the three states that were calculated from the simulation data after the agent obtained 500 reinforcers.**

broken-stick shape, which is an evidence that the underlying IRTs follow a bi-exponential distribution. Thus, only the dual model reproduced bout-and-pause patterns.

We posit that both of the choice and cost mechanisms are necessary to organize responses into bout-and-pause patterns. The no choice model failed because it lacks the choice mechanism. Without the choice mechanism, the agent almost always stayed in the Operant state and responded at a high rate without pauses. The reason behind the failure of the no cost model was the knockout of the cost mechanism. When the cost of a changeover is zero, the agent easily return to the Choice state, resulting in sporadic operant responses followed by long pauses. Similar behaviors were observed in pigeons under a concurrent VI VI schedule without COD [17]. The choice and cost mechanisms contribute differently to generate bout-and-pause patterns; the choice mechanism generates pauses and the cost mechanism produces response bursts. Since the dual model has both the mechanisms, it reproduced bout-and-pause patterns.

Since we have a full control of the simulation environment and the agent in it, we can exclude the possibility of contamination by other factors. Smith et al. [17]'s results implied that choice and cost are behind bout-and-pause patterns but it was not clear if other factors influence the formation of bout-and-pause patterns; this is an inherent limitation of experimental studies. It was not straightforward to draw conclusions like "these mechanisms are enough to

generate bout-and-pause patterns" from the experimental findings that IRT distributions observed in pigeons followed a bi-exponential distribution under concurrent VI VI schedules with a COD. In contrast, our constructive approach makes it clear that the two mechanisms are sufficient to reproduce bout-and-patterns, and this conclusion is hard to draw only from the experimental findings from [17].

We suggest that what is important for generating bout-and-pause patterns is not the specific architecture of our model but the choice and cost mechanisms. Our model is composed of three states and five equations, and those equations are from one of most popular reinforcement algorithms called Q-learning. Even if such model architecture and algorithm are substituted with others, the new model will still reproduce bout-and-pause patterns if it involves the choice and cost. The specific equation forms such as the logarithm in softmax function in Eq (3) or the logarithm in Eq (4) the can also be replaceable with other forms. We do not reject other possible forms to implement the two mechanisms.

We also do not claim the uniqueness of our experimental settings. Although we employed an FR 1 schedule for the other behaviors, other schedules including VI should produce similar results.

## 2 Simulation 2

Having demonstrated in Simulation 1 that the dual model successfully reproduced bout-and-pause patterns, in Simulation 2 we analyzed this model under various environments. The previous studies [4, 6–10, 27] have applied various experimental manipulations to animals to understand bout-and-pause patterns, as summarized in Table 1. We applied manipulations to the agent in the model by changing environmental settings.

### 2.1 Method: Simulation 2

Using the dual model, we performed four experiments by manipulating only one of the four variables while keeping the other three variables the same as Simulation 1. The procedure of simulation was also the same as Simulation 1.

The four experimental manipulations are applied independently to each of the four variables: 1) the rate of reinforcement, 2) the deprivation level, 3) the presence of extinction, and 4) the schedule type. 1) We manipulated the rate of reinforcement by varying mean intervals of the VI schedule. Mean intervals used in this simulation were VI 30 s, 120 s, and 480 s (2.0, 0.5, and 0.125 reinforcer per min). 2) We varied reward values obtained in the Operant state to control the deprivation level of the agent. Those values were 0.5, 1.0, and 1.5 to induce low deprivation, baseline, and high deprivation levels, respectively. The reward value that the agent received by taking other behaviors was the same as Simulation 1 throughout all the simulations. 3) To attenuate the engagement to the operant response, we switched the schedule from VI 120 s (0.5 reinforcer per min) to extinction after the agent obtained 1,000 reinforcers. The extinction phase finished when 3,600 s (36,000 time steps) elapsed. 4) We manipulated the schedule type by adding a small VR schedule in tandem to a variable time (VT) schedule. The mean interval of the VT schedule was fixed to 120 s and VR values were 0, 4, and 8.

When we analyzed the IRTs data from the extinction simulation, we used a dynamic bi-exponential model [10], in which the model parameters, $q$, $\omega$, and $b$, are time-dependent and Eq (1) is rewritten as follows:

$$p(IRT = \tau) = (1 - q_t)\omega_t e^{-\omega\tau} + q_t b_t e^{-b_t\tau}. \tag{6}$$

Extinction causes exponential decay of the model parameters according to the following equations:

$$1 - q_t \quad = (1 - q_0)e^{-\gamma t}, \tag{7}$$

$$b_t \quad = b_0 e^{-\delta t}, \tag{8}$$

where the parameters $\gamma$ and $\delta$ denote the decay rates of $q$ and $b$, respectively. Since the decay of any of the three model parameters $q$, $b$, and $\omega$ can cause extinction, we need to identify which of these parameters actually decayed during the extinction simulation. We excluded $\omega$ because it was fixed to 1/3 during the simulation. To identify whether one or both of the $q$ and/or $b$ parameters decayed, we compared three models, that is, the $qb$-decay, $q$-decay, and $b$-decay models. We calculated WAIC (widely applicable information criterion [28]) for each model. We use Markov chain Monte Carlo (MCMC) with Stan [29] to estimate posterior distribution and used MCMC samples to calculate WAIC. The same configuration as Simulation 1 was used in Simulation 2.

To examine the molar relationship between the reinforcement rate and response rate, we fitted Herrnstein's hyperbola [30] to the simulated data. We used its modern version [31],

$$R = \frac{kr^a}{r^a + r_e^a/c}, \tag{9}$$

where $R$ is the response rate, $r$ is the reinforcement rate, $r_e$ is the external reinforcement rate, $k$ is the total amount of behavior, and $a$ is the exponent and bias parameters, respectively. Since the parametrization of term $r_e^a/c$ is redundant, we did not fit $r_e$ and $c$ separately and estimated only $r_e^a/c$.

## 2.2 Results: Simulation 2

Fig 4 shows the log-survivor plots of IRTs from each of the four simulations. Fig 4(a) and 4(b) shows that manipulating the rate of reinforcement or the deprivation level changed the slope and intercept of the right limb. As the rate of reinforcement or the deprivation level increased, the slope of the right limb became steeper, indicating that the bout initiation rate became larger. The broken sticks in Fig 4(c) have different slopes and y-axis intercepts, suggesting that both the bout initiation rate and the bout length were changed. Fig 4(d) shows that adding the tandem VR schedule to the VT schedule affected only the y-axis intercept of the right limb without changing its slope. As the required response increased from the baseline to VR 4 or VR 8, the bout length became larger. However, the right limbs were not stable and we performed a fitting analysis described in the next paragraph.

Table 2 shows estimated parameters of the bi-exponential model, $q$, $\omega$, and $b$ in three simulations except for extinction. Parameter $q$ increased as the reinforcement rate, the deprivation level, and the number of required responses increased. Parameter $\omega$ did not change in all manipulations. Parameter $b$ increased as the rate of reinforcement and the deprivation level increased.

In Fig 4(c), the total number of IRTs during the extinction phase was insufficient to reliably estimate the right limb. Then, we analyzed the dynamic bi-exponential model fitted to the IRTs during extinction. Table 3 shows the WAIC values for the three models. The smallest WAIC was attained by the $qb$-decay model, but the differences from the other models are not large and it is not conclusive which of the bout initiation rate and the bout length decayed during extinction.

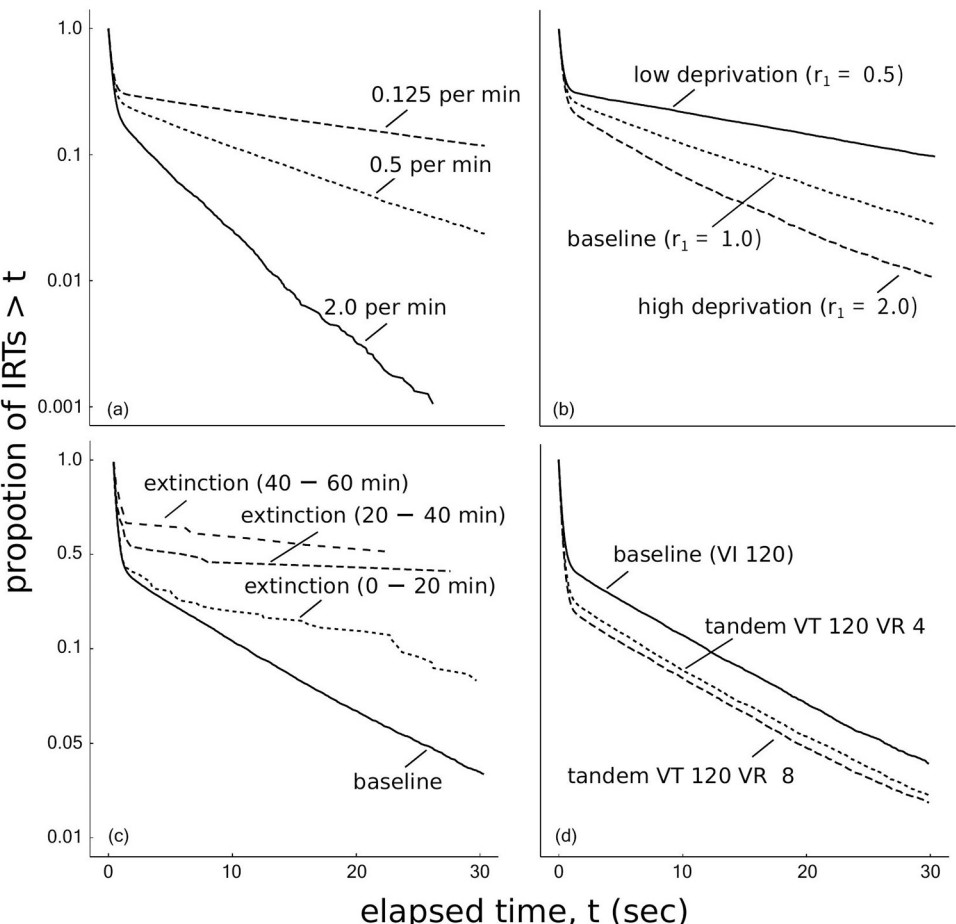

**Fig 4. The log-survivor plots of IRTs generated under the manipulation of (a) the rate of reinforcement, (b) the deprivation level, (c) the schedule type, and (d) the presence of extinction drawn by all data after the agent obtained 500 reinforcers.**

Fig 5 shows the boxplots of $Q_{\text{pref}}$ and $Q_{\text{cost}}$ in the three simulations except for extinction, which are to be used for assessing how the changes in the bout components are mediated. We excluded the extinction simulation because we already knew that $Q_{\text{pref}}$ causes the change of the bout components since $Q_{\text{cost}}$ is fixed during the extinction phase. The top panel shows that

**Table 2. Estimated parameters of the bi-exponential model in simulations.**

| Manipulation | Condition | $\omega$ | $b$ | $q$ |
|---|---|---|---|---|
| Rate of reinforcement | VI 30 (2.0 per min) | 3.08 | 0.23 | 0.17 |
| | VI 120 (0.5 per min) | 3.06 | 0.09 | 0.27 |
| | VI 480 (0.125 per min) | 3.19 | 0.03 | 0.31 |
| Deprivation level | High deprivation | 3.04 | 0.24 | 0.17 |
| | Baseline | 3.15 | 0.08 | 0.26 |
| | Low deprivation | 3.17 | 0.03 | 0.31 |
| Tandem VT 120 VR $x$ | VR 0 | 3.07 | 0.09 | 0.26 |
| | VR 4 | 3.23 | 0.08 | 0.19 |
| | VR 8 | 3.11 | 0.08 | 0.16 |

Table 3. Parameter selection for the dynamic bi-exponential model with WAIC.

| Model | WAIC |
|---|---|
| $qb$-decay | 1.936 |
| $b$-decay | 1.940 |
| $q$-decay | 1.980 |

The lower WAIC, the better the model.

$Q_{pref}$ and $Q_{cost}$ increased as the rate of reinforcement increased. The middle panel indicates that increasing the deprivation level moved $Q_{pref}$ and $Q_{cost}$ upward. From the bottom panel, we can see that adding tandem VR schedule increased $Q_{cost}$ without affecting $Q_{pref}$. Table 5 summarized the dependency of $Q_{pref}$ and $Q_{cost}$ to experimental manipulations. Comparing Tables 1 and 5, $Q_{pref}$ and $Q_{cost}$ correspondent to the bout initiation rate and the bout length.

Fig 6 shows the relationship between the reinforcement rate and response rate in our model. The response rate increased with diminished gradients, converging to $k = 187.41$. The other parameters were fitted to be $a = 2.25$, and $r_e^a/c = 2.65$. The percentage of variance accounted for (%VAF) was 99.3, and $a = 2.25$ implies that our model showed overmatching. In our model, $\beta$ in Eq (3) controls the absolute difference between the Operant and Others behaviors and we can change overmatching to strict matching by lowering the value of $\beta$.

## 2.3 Discussion: Simulation 2

In Simulation 2, we tested whether the dual model has the same characteristics as animals reported by the previous studies. We analyzed the model with four experimental manipulations: the rate of reinforcement, the deprivation level, the presence of extinction, and the schedule type. The rate of reinforcement, the deprivation level, and the presence of extinction affected the bout initiation rate and the bout length and adding the tandem VR schedule to the VT schedule affected only the bout length.

Table 4 summarizes the relationship between the experimental manipulations and the bout components observed in the dual model, which suggests that the behaviors of the dual model are consistent with the existing knowledge on animal behaviors. Furthermore, we made stable predictions to the cells with the question marks in Table 1. Our predictions are stable because our results can be easily reproduced and tested using the same simulation code. In contrast, experimental studies with animals could report different conclusions. Although our model does not implement Herrnstein's hyperbola a priori, the molar relationship between the reinforcement rate and response rate is well described by the modern matching theory (Fig 6). Cheung et al. [12] and Brackney et al. [32] showed that the bout initiation rate and the bout length decayed during extinction. Table 3 shows that parameter selection for dynamic bi-exponential model with WAIC but the differences between each model are small. However, the lowest WAIC model is consistent with previous studies. Therefore, the dual model satisfies at least the necessary conditions to be a model to be analyzed for the generation mechanism of bout-and-pause patterns.

Table 2 showed estimated parameters of the bi-exponential model in each simulation and they are consistent with the parameters of previous study with real animals.

The dependency of $Q_{pref}$ and $Q_{cost}$ to experimental manipulations, showed in Table 5, can be understood according to the categorization of motivational and schedule-type manipulations proposed by Shull et al. [4]. In our simulations, manipulating any of the three motivational variables, i.e. the rate of reinforcement, the deprivation level, or extinction, changed

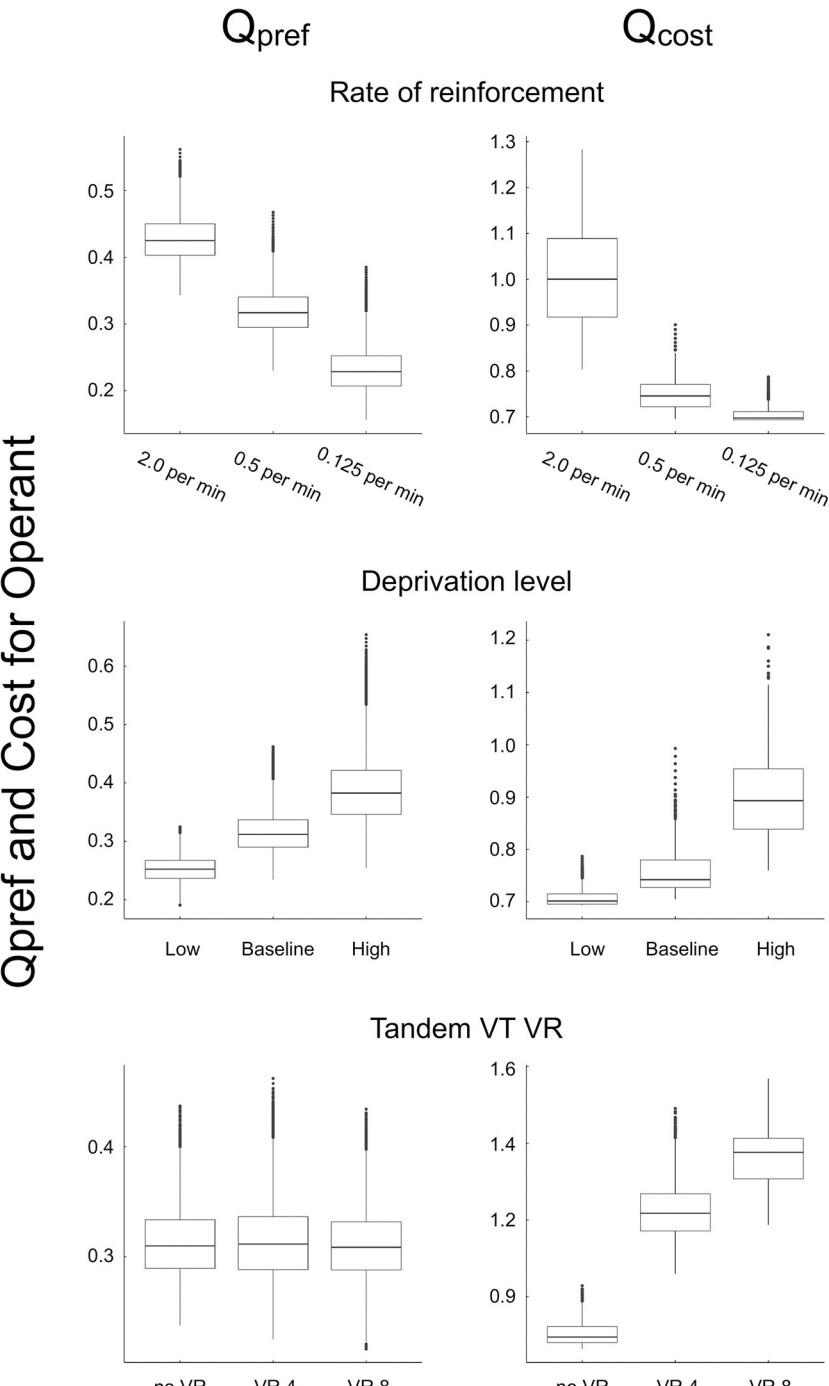

**Fig 5. Boxplots of $Q_{pref}$ and $Q_{cost}$ in each simulation.** The top, middle, and bottom rows correspond to the reinforcement rate, the deprivation level, and the tandem VT VR simulations, respectively, and the left and right columns show $Q_{pref}$ and $Q_{cost}$, respectively.

$Q_{pref}$ and $Q_{cost}$. The change of $Q_{cost}$ was not a primary but a secondary effect because $Q_{cost}$ was changed as a result of the increased $Q_{pref}$; with a higher $Q_{pref}$, the agent emits more responses. The schedule type manipulation affected only $Q_{cost}$. These changes of $Q_{pref}$ and $Q_{cost}$ are consistent with what was proposed by Shull et al. [4].

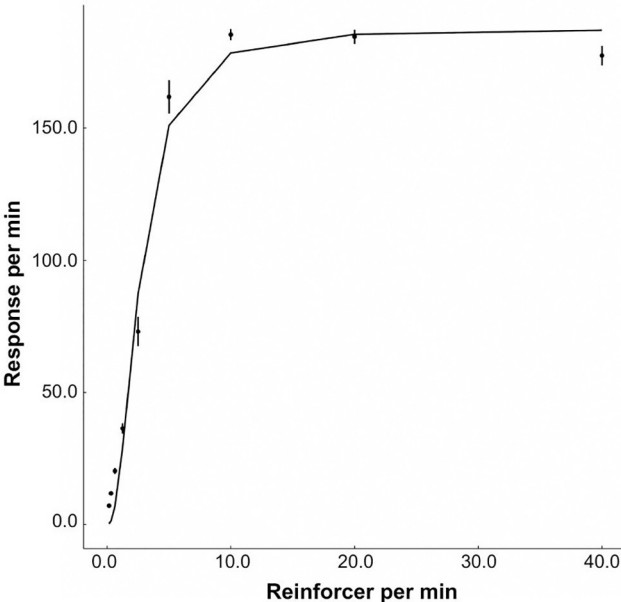

**Fig 6. The response rate as a function of the reinforcement rate.** The dots are from the simulation and the line is the modern version of Herrnstein's hyperbola (the generalized matching law) fitted to the data.

The dual model is limited to reproduce only some of the previous findings. Here are three examples of limitations. First, our model is not designed for analyzing the addition of a tandem VR schedule to a VT schedule, by which Tanno [9] and Matsui et al. [21] found the change of the within-bout response rate, which was fixed in our model. Second, the value and the delay between a response and a reinforcer were fixed in our model. Brackney et al. [10] and Podlesnik et al. [8] considered a delayed reinforcement from a bout initiation causes the inverse correlation between the bout initiation rate and the bout length. This result can be reproduced if required responses by tandem VR is very high (more than 32). Third, sometimes the bout length does not decrease during extinction [10]. Our dual model could not reproduce this result even if we changed the model parameters.

**Table 4. The behavior of the dual model.**

| | Motivational | | | Schedule type |
|---|---|---|---|---|
| | **Reinforcement rate** | **Deprivation level** | **Extinction** | **Tandem VR** |
| Bout length | ↗* | ↗* | ↘† | ↗* |
| Bout initiation rate | ↗* | ↗* | ↘* | →† |

The cells marked with "*" indicates the consistency with the animal findings shown in Table 1. The cells marked with "†" were the cells with "?" in Table 1.

**Table 5. The dependency of $Q_{\text{pref}}$ and $Q_{\text{cost}}$ to experimental manipulations in the dual model.**

| | Motivational | | | Schedule type |
|---|---|---|---|---|
| | **Reinforcement rate** | **Deprivation level** | **Extinction** | **Tandem VR** |
| $Q_{\text{cost}}$ | ↗ | ↗ | → | ↗ |
| $Q_{\text{pref}}$ | ↗ | ↗ | ↘ | → |

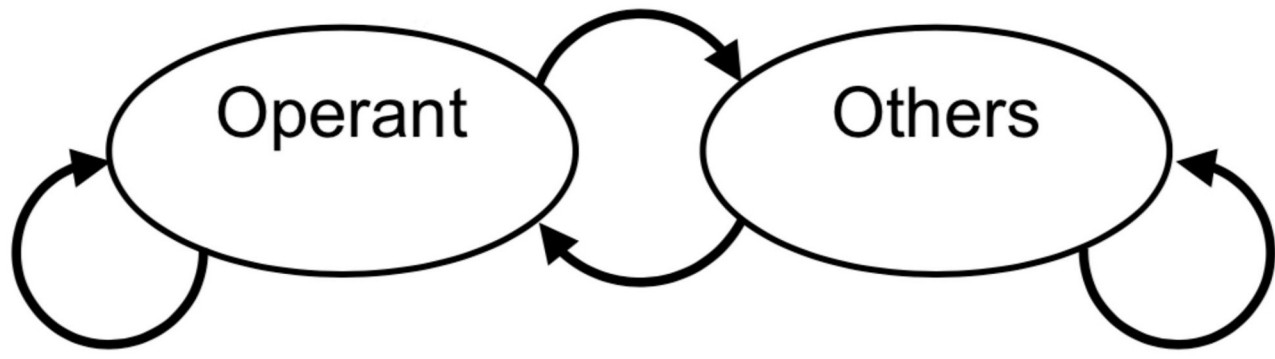

**Fig 7. Two-state model.**

## 3 Simulation 3

In Simulation 3, we examined a two-state model that incorporates the choice and cost mechanisms to examine the possibility of alternative models, particularly a simpler one. We built a two-state model without the Choice state and ran simulations with it.

### 3.1 Method: Simulation 3

Fig 7 shows the two-state model comprising of the Operant and Others states. Although it does not have the Choice state, the choice mechanism is implemented as the transitions between the Operant and the Others states. The probability of staying at the same state is defined as follows.

$$p(s_{t+1} = i | s_t = i) = \frac{\exp\left(w_{\mathrm{pref}} Q_{\mathrm{pref}}^{(i)}(t) + w_{\mathrm{cost}} Q_{\mathrm{cost}}^{(i)}(t)\right)}{\sum_i \exp\left(w_{\mathrm{pref}} Q_{\mathrm{pref}}^{(i)}(t) + w_{\mathrm{cost}} Q_{\mathrm{cost}}^{(i)}(t)\right)}, \tag{10}$$

where $w_{\mathrm{pref}}$ and $w_{\mathrm{cost}}$ are positive weights for $Q_{\mathrm{pref}}$ and $Q_{\mathrm{cost}}$, respectively. Updating rules for $Q_{\mathrm{pref}}$ and $Q_{\mathrm{cost}}$ are the same as Eqs (2a), (2b) and (4), respectively. The parameters of the two-state model were sought in the ranges shown in Table 6, which includes the parameter values used for the three-state, dual model. The following parameter settings were selected from the range: $\alpha_{\mathrm{rft}} = 0.01$, $\alpha_{\mathrm{ext}} = 0.01$, $w_{\mathrm{pref}} = 4.0$, $w_{\mathrm{cost}} = 3.5$, $r^{(\mathrm{Operant})} = 1.0$, and $r^{(\mathrm{Others})} = 0.5$.

To examine if the two-state model could generate bout-and-pause patterns and if it could be used for simulations with experimental manipulations, we performed simulation analysis. We varied the reinforcement rate as VI 30 s, VI 120 s, and VI 480 s, which were the same as the values used in Simulation 2.

### 3.2 Results: Simulation 3

Fig 8 shows the log-survivor plots of IRTs from the simulation of the two-state model with different values of the reinforcement rate. It showed broken-stick shapes and the slopes and intercepts of the right limbs decreased as the reinforcement rate decreased.

**Table 6. Parameter range of the two-state model.**

| Parameter | Min | Max | Step |
|:---:|:---:|:---:|:---:|
| $\alpha_{\mathrm{rft}}$ | 0.01 | 0.2 | 0.01 |
| $\alpha_{\mathrm{ext}}$ | 0.01 | 0.2 | 0.01 |
| $w_{\mathrm{pref}}$ | 1.0 | 6.0 | 0.1 |
| $w_{\mathrm{cost}}$ | 1.0 | 6.0 | 0.1 |

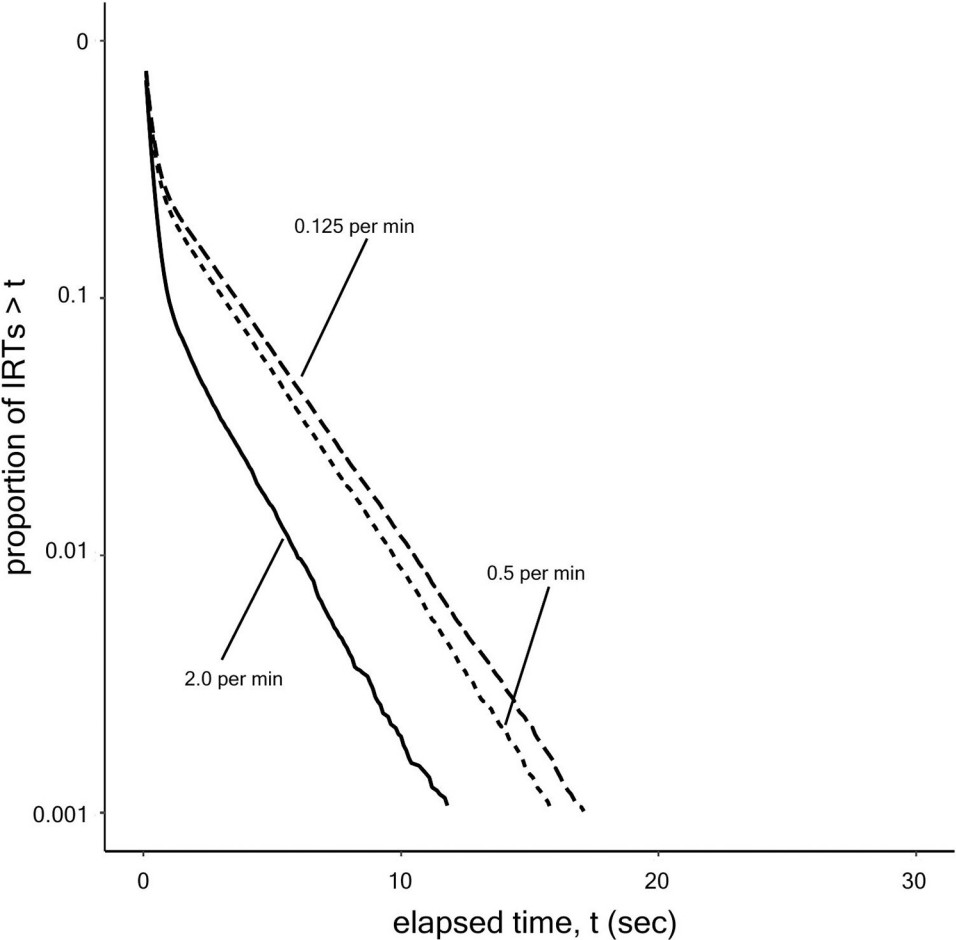

**Fig 8. Log-survivor plots of IRTs generated by the two-state model under VI 30 s, VI 120 s, and VI 480 s schedules.**

### 3.3 Discussion: Simulation 3

Since the log-survivor plots of IRTs generated by the two-state model showed broken-stick curves, bout-and-pause patterns were reproduced. In addition, the change of the log survivor plots of the two-state model was consistent with experimental findings. Therefore, we can construct alternative models even without the explicit third state. Also, the two-state model implements the two mechanisms through Eq (10).

We consider the three-state, dual model has advantages in modeling and analyzing bout-and-pause patterns.

In the three-state dual model, the effects of choice and cost are separated. It is clear in the dual model shown in Fig 1(a) that the choice between the operant and other behaviors is made at the Choice state and whether the agent continues to stay in the same state is moderated by the cost mechanism at each of the Operant and Others states. This can be understood by Eq (3), which describes only the choice rule, and Eq (5), which calculates the stay probability based on the cost mechanism. However, in the two-state model, choice and stay are not well separated; in Eq (10), choice and cost are mixed and the behavior of the agent cannot be explained by only one of them.

## 4 General discussion

In this paper, we have developed a computational model with reinforcement learning. The model was meant to explain how bout-and-pause patterns can be generated and we examined its validity by comparing computer simulations and experimental findings. We hypothesized that two independent mechanisms, the choice between Operant and Others and the cost in the changeover of behaviors, are necessary to organize responses into bout-and-pause patterns. We demonstrated in Simulation 1 that the dual model reproduced bout-and-pause patterns under a VI schedule. Simulation 2 found that the relationships between various experimental manipulations and the bout components in our model were consistent. Simulation 3 found that two-state model incorporating the two mechanisms can also reproduce the bout-and-pause pattern. However, the third state has advantages in analyzing an agent behavior because it separates the effects of the choice and cost mechanisms. These results support our hypothesis that assumes that an agent transitioning between the three states driven by the choice and cost mechanisms organizes its responses into bout-and-pause patterns. This is our answer to "why bout-and-pause patterns are organized?"

Our constructive model reproduced the descriptive results reported by [4, 6, 7, 27]. Although our dual model does not explicitly include the bi-exponential model in Eq (1), IRTs generated by the dual model followed the bi-exponential model.

The fundamental difference between our model based on reinforcement learning and Kulubekova and McDowell [22]'s model based on selection by consequences is that our model explicitly has the choice and cost mechanisms but Kulubekova and McDowell [22]'s model is unclear about them. Their model did not generate a clear distinction between a burst of responses in a short period and long pauses that separate bursts, resulting in a dull bend of the log-survivor plot. Kulubekova and McDowell [22] discussed that this divergence from live animals might be due to the lack of CODs in their model. Our model reproduced clear distinction between bursts and pauses (Fig 2), and this was because our model can change CODs through the cost mechanism. Another advantage of us over Kulubekova and McDowell [22] is that they did not compare with alternative models but we tested our hypothesises of the choice and cost mechanisms by the knockout analysis Simulation 1.

Our model has at least two shortcomings, which are the range of parameters and the redundancy of the model. The parameters $\alpha_{\text{rft}}$, $\alpha_{\text{ext}}$, $\beta$, $\omega_{\text{pref}}$, and $\omega_{\text{cost}}$ in our model have not been optimized to fit to behavioral data from real animals. The evidence that supports our parameter selection is that our model quantitatively reproduced bout-and-pause patterns. Second, although our model has five parameters, fewer parameters may suffice to reproduce bout-and-pause patterns. To verify our model for these two points, it would be useful to compare empirical data from real animals and our computational model.

Standing on our model proposed in this paper, we can extend our research to many directions to explain more various aspects of bout-and-pause patterns. Here we discuss four of them. The following four paragraphs are devoted to this topic.

First, our results were retrospective to data from previous behavioral experiments and the proposed model was not tested by its prediction ability to unseen data. Our model can suggest a new experiment that could add new knowledge about how manipulating CODs affect animals' behavior under a concurrent VI VI schedule. Smith et al. [17] pointed out that employing asymmetrical CODs in a concurrent VI VI schedule could produce behaviors under a single response schedule. Our modeling is consistent to what Smith et al. [17] pointed out but approaches from a different direction. We consider that, even if an animal is under the single schedule, it makes choices between the operant behavior and other behaviors; this is implemented to our simulation as concurrent VI FR 1. We used an FR 1 schedule for other

behaviors in our simulations, but we can change it from FR 1 to a VI schedule so that the whole schedule becomes a concurrent VI VI schedule. In our model, the cost for the operant behavior, defined by $Q_{\text{cost}}^{(\text{Operant})}$, affects actions of the agent only in the Operant state, without affecting those in the Others state; similarly, the cost for other behaviors influences the agent to its actions only in the Others state. Therefore, according to our model, it is expected that, in a concurrent VI VI schedule, if the experimenter varies CODs for one schedule, the behavior of the animal changes only for the varied schedule without affecting the behavior for the other schedule. It will be interesting to conduct such experiments with real animals to reveal actual effects of CODs on the behavior under concurrent VI VI schedules. In such a way, our model can bridge between animal behaviors observed in concurrent schedules and single schedules by offering a unified framework.

The second approach is verification based on neuroscientific knowledge. Even if the model can correctly predict unseen data from behavioral experiments, it is not guaranteed that animals employ the same the model. To explore real mechanisms that animals implement, it would be effective to compare the internal variables of the model with neural activities measured from real animals during behavioral experiments. Possible experiments are to perform knock out experiments by inducing lesions at specific areas of the brain that should be active during the experiments, or to activate or deactivate specific neurons during the experiment.

Third, we can assess the plausibility of our model in more detail by conducting simulation under new experimental manipulations including disruptors or analyzing measures that we did not analyze. For example, recent studies showed that the distribution of bout lengths is sensitive to experimental manipulations [13, 33, 34]. Sanabria et al. [35] have proposed a computational formulation of behavior systems [36] and their descriptive model well described bout-and-pause patterns including the distribution of bout lengths.

Fourth, we can design models that are not Markov transition models. The bout-and-pause response patterns shown in Fig 2 can be generated by a Markov transition model whose transition matrix is given a priori without reinforcement learning. We argue that the statistical description of the Markov model (i.e., the transition matrix defined by the transition probabilities shown in Fig 3) is not the source of the reproducibility of bout-and-pause patterns. There may be other models that are not formulated by Markov transition, such as the model proposed by McDowell [23]. We can introduce the choice and cost mechanisms to such models.

Reinforcement learning can be employed to model and explain animal behaviors other than bout-and-pause patterns, since it is a general framework where an agent learns optimal behaviors in a given environment through trial-and-error [24]. Such a reinforcement learning framework agrees well with the three-term contingency in behavior analysis. There are three essential elements in reinforcement learning; a state, an action, and a reward. The state is what agent observe and is information about the environment. The action is a behavior that the agent takes in a given state. The reward is what the agent obtains as the result of the action. These three elements are similar to a discriminative stimulus, a response, and an outcome. This similarity would allow behavior analysts to employ reinforcement learning in their research. For example, Sakai and Fukai [37] employed actor-critic reinforcement learning to modeling the matching law. We hope more computational studies will be performed to expand methods of behavioral science.

## Author Contributions

**Conceptualization:** Kota Yamada.

**Formal analysis:** Kota Yamada, Atsunori Kanemura.

**Funding acquisition:** Kota Yamada, Atsunori Kanemura.

**Investigation:** Kota Yamada.

**Methodology:** Kota Yamada.

**Software:** Kota Yamada.

**Supervision:** Atsunori Kanemura.

**Visualization:** Kota Yamada, Atsunori Kanemura.

**Writing – original draft:** Kota Yamada, Atsunori Kanemura.

**Writing – review & editing:** Kota Yamada, Atsunori Kanemura.

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
