## [Decision Letter · Decision Letter 0]

1 Sep 2020

PONE-D-20-18275

Simulating bout-and-pause patterns with reinforcement learning

PLOS ONE

Dear Dr. Yamada,

Thank you for submitting your manuscript to PLOS ONE. After careful consideration, we feel that it has merit but does not fully meet PLOS ONE’s publication criteria as it currently stands. Therefore, we invite you to submit a revised version of the manuscript that addresses the points raised during the review process.

The manuscript should be revised for clarity of the presentation and appropriate references should be discussed and appropriately acknowledged.

Please, provide a  statistical description of the  three-state Markov model. The transition matrix could be analytically computed in at least some limit cases. That could give the authors some indications regarding the limitations of their model (besides the limitations mentioned in the general discussion of their manuscript).

Reviewers raised concerns about the embedded (hidden) assumptions in the computational model that lead to “emergent” properties. For example, the authors mentioned that “Although our dual model does not explicitly include the bi-exponential model in Eq. (1), IRTs generated by the dual model followed the bi-exponential model.” Please, discuss a possibility that the “natural” assumption of a Boltzmann factor in Eq.3 of the model or the logarithmic formula in Eq. 4 does not lead to the “emergent” behavior, such as the bi-exponential model form Eq. 1? How do the authors know that this simple assumption is not the root of the observed bi-exponential and other exciting features?

Pleaser, discuss the potential limitations of the model.

Some  important recent papers are missing from discussion and should be included in the revision.

Brackney, R. J., Cheung, T. H. C., & Sanabria, F. (2017). A bout analysis of operant response disruption. Behavioural Processes, 141(Part 1). https://doi.org/10.1016/j.beproc.2017.04.008

Brackney, R. J., & Sanabria, F. (2015). The distribution of response bout lengths and its sensitivity to differential reinforcement. Journal of the Experimental Analysis of Behavior, 104(2), 167–185. https://doi.org/10.1002/jeab.168

Chen, X., & Reed, P. (2020). Factors controlling the micro-structure of human free-operant behaviour: Bout-initiation and within-bout responses are effected by different aspects of the schedule. Behavioural Processes, 175(March), 104106. https://doi.org/10.1016/j.beproc.2020.104106

Daniels, C. W., & Sanabria, F. (2017). About bouts: A heterogeneous tandem schedule of reinforcement reveals dissociable components of operant behavior in Fischer rats. Journal of Experimental Psychology: Animal Learning and Cognition, 43(3), 280–294. https://doi.org/10.1037/xan0000144

Jiménez, Á. A., Sanabria, F., & Cabrera, F. (2017). The effect of lever height on the microstructure of operant behavior. Behavioural Processes, 140, 181–189. https://doi.org/10.1016/j.beproc.2017.05.002

Reed, P. (2015). The structure of random ratio responding in humans. Journal of Experimental Psychology: Animal Learning and Cognition, 41(4), 419–431.

Reed, P., Smale, D., Owens, D., & Freegard, G. (2018). Human performance on random interval schedules. Journal of Experimental Psychology: Animal Learning and Cognition, 44(3), 309–321.

Sanabria, F., Daniels, C. W., Gupta, T., & Santos, C. (2019). A computational formulation of the behavior systems account of the temporal organization of motivated behavior. Behavioural Processes, 169, 103952. https://doi.org/10.1016/j.beproc.2019.103952

Brackney et al. (2017), for instance, report on the effect of various disruptors, including extinction, on bout-organized behavior. Although the distribution of bout lengths is not assessed in the proposed model, it may be important to note that research on that front has been conducted (Brackney & Sanabria, 2015; Jiménez et al., 2017). Also, the proposed model is, in some aspects, comparable to the partially hidden Markov model proposed by Sanabria et al. (2019)—the latter is not a learning model, but accounts for stable-state bi-exponential distribution of IRTs without building that distribution in the model itself.

In page 2, it should be pointed out that the bi-exponential distribution of IRTs has been demonstrated in VI schedules, where reinforcement is available probabilistically at a constant rate. Later in the manuscript the authors make reference to the schedules of reinforcement without explaining them. Also, q does not correspond to the length of a bout but to the *mean* length of a bout.

In page 3, the authors generalize the results from pigeons in Smith et al. (2014) to all animals, when rats actually show a very different pattern. The conclusion they reach is reasonable, assuming that rats engage in alternative behaviors during conditioning.

Page 4, line 5: “both of” should be “both”

Figure 1: Please use a larger font size.

Line 129: “knowledge that is observed” is a strange, ambiguous expression.

Line 170: Do you mean “Fechner’s law”, which implies a representation of magnitude (here, number of lever presses) in logarithmic space. Weber’s law does not imply such representation.

Line 242: “We posit both…” should be “We posit that both…”

Equation 9: Its description includes a parameter b that is not included in the equation.

Line 487: “real animals may have fewer parameters” is a strange expression.v

We look forward to receiving your revised manuscript.

Kind regards,

Gennady Cymbalyuk, Ph.D.

Academic Editor

PLOS ONE

Journal Requirements:

2. Thank you for including your competing interests statement; "The authors have no competing interests."

We note that one or more of the authors are employed by a commercial company: LeapMind Inc.

3. We note you have included a table to which you do not refer in the text of your manuscript. Please ensure that you refer to Table 5 in your text; if accepted, production will need this reference to link the reader to the Table.

Reviewers' comments:

Reviewer's Responses to Questions

**Comments to the Author**

1. Is the manuscript technically sound, and do the data support the conclusions?

Reviewer #1: Yes

Reviewer #2: Yes

2. Has the statistical analysis been performed appropriately and rigorously? 

Reviewer #1: Yes

Reviewer #2: Yes

3. Have the authors made all data underlying the findings in their manuscript fully available?

Reviewer #1: Yes

Reviewer #2: Yes

4. Is the manuscript presented in an intelligible fashion and written in standard English?

Reviewer #1: Yes

Reviewer #2: Yes

5. Review Comments to the Author

Reviewer #1: Yamada and Kanemura propose a parsimonious yet insightful reinforcement learning model that successfully reproduces the bout-like temporal organization of instrumental behavior. Moreover, the model reasonably links key experimental manipulations to parameters of the model. In its most complex version, the model is a 3-state (choice, operant, other) Markov chain with updatable transition probabilities. The authors judiciously attempt to simplify the model further, showing that such simplifications come at great heuristic cost. It is particularly commendable that the authors acknowledge the potential limitations of the model.

I only have two relatively minor concerns regarding the manuscript. First, although the authors provide a useful synthesis of the literature on the microstructure of instrumental behavior, many important recent papers are missing from that synthesis, which makes it appear outdated. Below I have listed several recent papers that the authors omitted, and that I believe would inform the assessment of their model. Because I am co-author in many of these papers, I am disclosing my name in the signature, and my recommendation will not change whether or not the authors choose to include any of them.

Brackney, R. J., Cheung, T. H. C., & Sanabria, F. (2017). A bout analysis of operant response disruption. Behavioural Processes, 141(Part 1). https://doi.org/10.1016/j.beproc.2017.04.008

Brackney, R. J., & Sanabria, F. (2015). The distribution of response bout lengths and its sensitivity to differential reinforcement. Journal of the Experimental Analysis of Behavior, 104(2), 167–185. https://doi.org/10.1002/jeab.168

Chen, X., & Reed, P. (2020). Factors controlling the micro-structure of human free-operant behaviour: Bout-initiation and within-bout responses are effected by different aspects of the schedule. Behavioural Processes, 175(March), 104106. https://doi.org/10.1016/j.beproc.2020.104106

Daniels, C. W., & Sanabria, F. (2017). About bouts: A heterogeneous tandem schedule of reinforcement reveals dissociable components of operant behavior in Fischer rats. Journal of Experimental Psychology: Animal Learning and Cognition, 43(3), 280–294. https://doi.org/10.1037/xan0000144

Jiménez, Á. A., Sanabria, F., & Cabrera, F. (2017). The effect of lever height on the microstructure of operant behavior. Behavioural Processes, 140, 181–189. https://doi.org/10.1016/j.beproc.2017.05.002

Reed, P. (2015). The structure of random ratio responding in humans. Journal of Experimental Psychology: Animal Learning and Cognition, 41(4), 419–431.

Reed, P., Smale, D., Owens, D., & Freegard, G. (2018). Human performance on random interval schedules. Journal of Experimental Psychology: Animal Learning and Cognition, 44(3), 309–321.

Sanabria, F., Daniels, C. W., Gupta, T., & Santos, C. (2019). A computational formulation of the behavior systems account of the temporal organization of motivated behavior. Behavioural Processes, 169, 103952. https://doi.org/10.1016/j.beproc.2019.103952

Brackney et al. (2017), for instance, report on the effect of various disruptors, including extinction, on bout-organized behavior. Although the distribution of bout lengths is not assessed in the proposed model, it may be important to note that research on that front has been conducted (Brackney & Sanabria, 2015; Jiménez et al., 2017). Also, the proposed model is, in some aspects, comparable to the partially hidden Markov model proposed by Sanabria et al. (2019)—the latter is not a learning model, but accounts for stable-state bi-exponential distribution of IRTs without building that distribution in the model itself.

The second concern is about style—not nearly as important as content, which is excellent in this paper, but it is important nonetheless. In various parts, the manuscript would benefit from economy of expression, precision, clearer organization of key claims in separate paragraphs, and a more deliberately logical connection between ideas. Below I just point at some salient examples:

In page 2, it should be pointed out that the bi-exponential distribution of IRTs has been demonstrated in VI schedules, where reinforcement is available probabilistically at a constant rate. Later in the manuscript the authors make reference to the schedules of reinforcement without explaining them. Also, q does not correspond to the length of a bout but to the *mean* length of a bout.

In page 3, the authors generalize the results from pigeons in Smith et al. (2014) to all animals, when rats actually show a very different pattern. The conclusion they reach is reasonable, assuming that rats engage in alternative behaviors during conditioning.

Page 4, line 5: “both of” should be “both”

Figure 1: Please use a larger font size.

Line 129: “knowledge that is observed” is a strange, ambiguous expression.

Line 170: I believe the authors mean “Fechner’s law”, which implies a representation of magnitude (here, number of lever presses) in logarithmic space. Weber’s law does not imply such representation.

Line 242: “We posit both…” should be “We posit that both…”

Equation 9: Its description includes a parameter b that is not included in the equation.

Line 487: “real animals may have fewer parameters” is a strange expression.

Federico Sanabria

Associate Professor of Psychology

Arizona State University

Reviewer #2: The manuscript expands on the previous work of Kota Yamada (see reference 15, where they analyzed the statistics of within–bout and bout-initiation). This work, in particular, is inspired by the research done in McDowell’s lab at Emory.

Briefly, the beauty of the model is its parsimony. The authors considered that the bout-and-pause patterns could be captured by a three-state Markov model controlled by two independent mechanisms: (1) the choice between Operant and Others, and (2) the cost in the changeover of behaviors.

At the same time, a three-state Markov is amenable to at least a basic statistical description, and the authors did not attempt that. The transition matrix could be analytically computed in at least some limit cases. That could give the authors some indications regarding the limitations of their model (besides the limitations mentioned in the general discussion of their manuscript).

The second concern I have is about the embedded (hidden) assumptions in the computational model that lead to “emergent” properties. For example, the authors mentioned that “Although our dual model does not explicitly include the bi-exponential model in Eq. (1), IRTs generated by the dual model followed the bi-exponential model.” My question is: how do they know that the “natural” assumption of a Boltzmann factor in Eq.3 of the model or the logarithmic formula in Eq. 4 does not lead to the “emergent” behavior, such as the bi-exponential model form Eq. 1? I understand that everybody used Boltzmann’s factor in every field of science, but still – how do the authors know that this simple assumption is not the root of the observed bi-exponential and other exciting features?

I also understand that both of my concerns are hard to address, but maybe the authors could at least comment on how they would address them.

6. PLOS authors have the option to publish the peer review history of their article (what does this mean?). If published, this will include your full peer review and any attached files.

Reviewer #1: **Yes: **Federico Sanabria

Reviewer #2: **Yes: **Sorinel Oprisan

---

## [Author Response · Author response to Decision Letter 0]

4 Oct 2020

Many thanks for considering our manuscript for possible publication in PLOS ONE. We have revised our manuscript based on the editor and the reviwers's comments. We believe our manuscript has significantly been improved thanks to comments from the editors and the reviwers.

Editor's Comment 1: Please, provide a statistical description of the three-state Markov model. The transition matrix could be analytically computed in at least some limit cases. That could give the authors some indications regarding the limitations of their model (besides the limitations mentioned in the general discussion of their manuscript).

In the General Discussion section, we have added a paragraph discussing that the statistical description (i.e., the transition probabilities shown in Figure 3) of the three-state Markov model gives an indication on the limitations of our model. The second paragraph from the last of the revised manuscript reads:

" Fourth, we can design models that are not Markov transition models. The bout-and-pause response patterns shown in Fig. 2 can be generated by a Markov transition model whose transition matrix is given a priori without reinforcement learning. We argue that the statistical description of the Markov model (i.e., the transition matrix defined by the transition probabilities shown in Fig. 3) is not the source of the reproducibility of bout-and-pause

patterns. There may be other models that are not formulated by Markov transition, such as

the model proposed by McDowell [23]. We can introduce the choice and cost mechanismssuch models. "

Editor's Comment 2: Reviewers raised concerns about the embedded (hidden) assumptions in the computational model that lead to “emergent” properties. For example, the authors mentioned that “Although our dual model does not explicitly include the bi-exponential model in Eq. (1), IRTs generated by the dual model followed the bi-exponential model.” Please, discuss a possibility that the “natural” assumption of a Boltzmann factor in Eq.3 of the model or the logarithmic formula in Eq. 4 does not lead to the “emergent” behavior, such as the bi-exponential model form Eq. 1? How do the authors know that this simple assumption is not the root of the observed bi-exponential and other exciting features?

Thank you for raising the question that how we know the specific form equation used in our model is not the cause of bout-and-patterns. To answer this question, we conducted a simulation with a modified model, where the Boltzmann factor and the logarithmic formula were replaced as follows.

●Use the matching law pi = Qi / ∑ Qi instead of the Boltzmann-type softmax function.

●Use square root instead of logarithm.

The result is shown in Fig. R1 below, which looks similar to Fig. 4(a). It implies that specific forms of equations such as the Boltzman factor in Eq. (3) and the logarithm in Eq. (4) arethe cause of bout-and-pause patterns. The second-to-last paragraph of the "Discussion of Simulation 1 section now has a new sentence: 

" The specific equation forms such as the softmax function Eq. (3) or the logarithm in Eq. (4) can also be replaceable with other forms."

Editor's Comment 3: Pleaser, discuss the potential limitations of the model.

As described in our response to Editor's Comment 1, we have added discussion on the limitation and extendability of the model. The concern raised in Editor's Comment 2 on specific equation forms was found not to be a fundamental limitation of the model since replacing the specific forms did not change the simulation results.

Editor's Comment 4: Some important recent papers are missing from discussion and should be included in the revision.

Thank you for enumerating recent important papers we missed in the previous manuscript. We reffered all of them from appropriate locations of the revised manuscript.

Editor's Comment 5: Brackney et al. (2017), for instance, report on the effect of various disruptors, including extinction, on bout-organized behavior. Although the distribution of bout lengths is not assessed in the proposed model, it may be important to note that research on that front has been conducted (Brackney & Sanabria, 2015; Jiménez et al., 2017). Also, the proposed model is, in some aspects, comparable to the partially hidden Markov model proposed by Sanabria et al. (2019)—the latter is not a learning model, but accounts for stable-state bi-exponential distribution of IRTs without building that distribution in the model itself.

Thank you for pointing out the importance of the research front issues. We have added a discussion on this point to General Discussion in our revised manuscript. The second-to-last paragraph of General Discussion of the revised manuscript reads: " Third, we can assess the plausibility of our model in more detail by conducting simulation under new experimental manipulations including disruptors or analyzing measures that we did not analyze. For example, recent studies showed that the distribution of bout lengths is sensitive to experimental manipulations [13, 33, 34]. Sanabria et al. [35] have proposed a computational formulation of behavior systems [36] and their descriptive model well described bout-and-pause patterns including the distribution of bout lengths. "

Editor’s Comment 6: In page 2, it should be pointed out that the bi-exponential distribution of IRTs has been demonstrated in VI schedules, where reinforcement is available probabilistically at a constant rate. Later in the manuscript the authors make reference to the schedules of reinforcement without explaining them. Also, q does not correspond to the length of a bout but to the *mean* length of a bout.

Thank you for pointing out our insufficiencies on the explanation about VI schedules. In the third paragraph of Introduction, we added a brief description of VI schedules and specified bout-and-pause patterns are observed under this schedule. Also, we have inserted "mean" to the description of  q .

Editor's Comment 7: In page 3, the authors generalize the results from pigeons in Smith et al. (2014) to all animals, when rats actually show a very different pattern. The conclusion they reach is reasonable, assuming that rats engage in alternative behaviors during conditioning.

In the sixth paragraph of Introduction, we specified that rats also engage alternative behaviors (i.e. schedule induced behavior, interim behavior, or adjunctive behavior) during conditioning. The added sentence is: " Similar observations have been made for rats, assuming that they engage in alternative behaviors during conditioning [20]. "

Editor’s Comment 8:

Page 4, line 5: “both of” should be “both”

Figure 1: Please use a larger font size.

Line 129: “knowledge that is observed” is a strange, ambiguous expression.

Line 170: Do you mean “Fechner’s law”, which implies a representation of magnitude

(here, number of lever presses) in logarithmic space. Weber’s law does not imply

such representation.

Line 242: “We posit both...” should be “We posit that both...”

Equation 9: Its description includes a parameter b that is not included in the equation.

Line 487: “real animals may have fewer parameters” is a strange expression.

We have fixed all of these points.

---

## [Decision Letter · Decision Letter 1]

29 Oct 2020

Simulating bout-and-pause patterns with reinforcement learning

PONE-D-20-18275R1

Dear Dr. Yamada,

We’re pleased to inform you that your manuscript has been judged scientifically suitable for publication and will be formally accepted for publication once it meets all outstanding technical requirements.

Kind regards,

Gennady Cymbalyuk, Ph.D.

Academic Editor

PLOS ONE

Additional Editor Comments (optional):

Reviewers' comments:

Reviewer's Responses to Questions

**Comments to the Author**

1. If the authors have adequately addressed your comments raised in a previous round of review and you feel that this manuscript is now acceptable for publication, you may indicate that here to bypass the “Comments to the Author” section, enter your conflict of interest statement in the “Confidential to Editor” section, and submit your "Accept" recommendation.

Reviewer #1: All comments have been addressed

Reviewer #2: All comments have been addressed

2. Is the manuscript technically sound, and do the data support the conclusions?

Reviewer #1: Yes

Reviewer #2: Yes

3. Has the statistical analysis been performed appropriately and rigorously? 

Reviewer #1: Yes

Reviewer #2: Yes

4. Have the authors made all data underlying the findings in their manuscript fully available?

Reviewer #1: Yes

Reviewer #2: Yes

5. Is the manuscript presented in an intelligible fashion and written in standard English?

Reviewer #1: Yes

Reviewer #2: Yes

6. Review Comments to the Author

Reviewer #1: The revised version of the manuscript addresses all concerns raised in the previous review. I have no further comments.

Reviewer #2: The authors attempted answering my questions the best they could. I understand that a more detailed answer than what few phrases they provided would actually mean adding a new section to the paper, which probably thye don't want at this stage.

7. PLOS authors have the option to publish the peer review history of their article (what does this mean?). If published, this will include your full peer review and any attached files.

Reviewer #1: **Yes: **Federico Sanabria

Reviewer #2: **Yes: **Sorinel A Oprisan

---

## [Editor Report · Acceptance letter]

3 Nov 2020

PONE-D-20-18275R1 

Simulating bout-and-pause patterns with reinforcementlearning 

Dear Dr. Yamada:

I'm pleased to inform you that your manuscript has been deemed suitable for publication in PLOS ONE. Congratulations! Your manuscript is now with our production department. 

Kind regards, 

on behalf of

Dr. Gennady Cymbalyuk 

Academic Editor

PLOS ONE